# The Impact of Mouse Preterm Birth Induction by RU-486 on Microglial Activation and Subsequent Hypomyelination

**DOI:** 10.3390/ijms23094867

**Published:** 2022-04-27

**Authors:** Cécile Morin, David Guenoun, Irvin Sautet, Valérie Faivre, Zsolt Csaba, Leslie Schwendimann, Pierrette Young-Ten, Juliette Van Steenwinckel, Pierre Gressens, Cindy Bokobza

**Affiliations:** 1NeuroDiderot, Inserm UMR-1141, Hôpital Robert Debré, Université de Paris, 75019 Paris, France; cecile.morin@inserm.fr (C.M.); david.guenoun@inserm.fr (D.G.); irvin.sautet@inserm.fr (I.S.); valerie.faivre@inserm.fr (V.F.); zsolt.csaba@inserm.fr (Z.C.); leslie.schwendimann@inserm.fr (L.S.); pierrette.young-ten@inserm.fr (P.Y.-T.); juliette.van-steenwinckel@inserm.fr (J.V.S.); pierre.gressens@inserm.fr (P.G.); 2AP-HP, Hôpital Robert Debré, 75019 Paris, France

**Keywords:** preterm birth, induced labor, mifepristone, RU-486, microglia, oligodendrocyte maturation, myelinization

## Abstract

Preterm birth (PTB) represents 15 million births every year worldwide and is frequently associated with maternal/fetal infections and inflammation, inducing neuroinflammation. This neuroinflammation is mediated by microglial cells, which are brain-resident macrophages that release cytotoxic molecules that block oligodendrocyte differentiation, leading to hypomyelination. Some preterm survivors can face lifetime motor and/or cognitive disabilities linked to periventricular white matter injuries (PWMIs). There is currently no recommendation concerning the mode of delivery in the case of PTB and its impact on brain development. Many animal models of induced-PTB based on LPS injections exist, but with a low survival rate. There is a lack of information regarding clinically used pharmacological substances to induce PTB and their consequences on brain development. Mifepristone (RU-486) is a drug used clinically to induce preterm labor. This study aims to elaborate and characterize a new model of induced-PTB and PWMIs by the gestational injection of RU-486 and the perinatal injection of pups with IL-1beta. A RU-486 single subcutaneous (s.c.) injection at embryonic day (E)18.5 induced PTB at E19.5 in pregnant OF1 mice. All pups were born alive and were adopted directly after birth. IL-1beta was injected intraperitoneally from postnatal day (P)1 to P5. Animals exposed to both RU-486 and IL-1beta demonstrated microglial reactivity and subsequent PWMIs. In conclusion, the s.c. administration of RU-486 induced labor within 24 h with a high survival rate for pups. In the context of perinatal inflammation, RU-486 labor induction significantly decreases microglial reactivity in vivo but did not prevent subsequent PWMIs.

## 1. Introduction

Preterm birth (PTB) represents 15 million births every year worldwide. In the preterm infant population, the neurodevelopmental consequences of being born too early are heterogeneous and strongly influenced by the gestational age when delivery occurs [1]. Preterm birth survivors suffer from white matter lesions [2], which are linked to neurodevelopmental disorders (NDDs), including attention deficit and hyperactivity disorder (ADHD) and autism spectrum disorder (ASD) [3,4]. Oligodendrocytes (OLs) are myelin protein producers within the brain parenchyma. In the case of preterm birth, an arrest of their differentiation at OL precursors (OPCs) occurs. OPCs are not able to produce myelin proteins; thus, a delay in brain myelinization (also known as hypomyelination) occurs, leading to periventricular white matter injuries (PWMIs) [5,6].

One of the main pathophysiological mechanisms described with respect to these neurological lesions is neuroinflammation induced by systemic inflammation [7]. Microglia, as the resident immunologic cells in the central nervous system (CNS), are the main actors that respond to environmental *stimuli*, including inflammatory signals [8]. Microglia reactivity is described in the literature in several brain pathologies, both in pediatric patients and adults [9,10]. In post-mortem tissue from preterm infants, microglia are activated [11]. Microglia develop and differentiate in the white matter at 4.5 gestational weeks before colonizing all the brain parenchyma [12]. Beyond their main immune function, microglia are fundamental to normal brain functioning during development by promoting myelination and the differentiation of neuronal progenitors, and by participating in the elaboration of the brain connectivity network through synaptic pruning [13]. In a mouse model characterized by interleukin (IL)-1β injections during the five first days of life, it is possible to recapitulate some key elements of PWMIs [6]. Previously, our team demonstrated in this model that (i) microglia disengage from their physiological transcriptomic profile [14], and (ii) reactive microglia are directly responsible for hypomyelination leading to PWMIs [15,16].

Medically, physicians consider three delivery modes: spontaneous vaginal delivery, vaginal delivery after labor induction, and cesarean section delivery. Currently, the literature lacks information concerning how the delivery mode may impact brain development. This leads to an absence of recommendations concerning the delivery mode in PTB cases. Based on the evidence that early infections, such as intrauterine infection, contribute to 25% to 40% of all human preterm births [17], animal models of early induced vaginal delivery using intraperitoneal or intrauterine injections of infection elements were developed (reviewed in [18]). Mice injected with *Escherichia coli (E. coli)* or lipopolysaccharide (LPS) on embryologic day (E)14.5–15.5 were delivered within 7 to 48 h [19,20]. Inflammation without infection can also lead to preterm birth: an increased level of IL-1-cytokine has been described in the uteruses of women who experience premature delivery [21]. A model of IL-1β injection in mice between E15 and E17 provoked delivery within 24 h [22]. These studies described various rates of pup survival. Many animal models of induced PTB already exist, but there is a lack of information regarding clinically used pharmacological substances to induce PTB and their consequences on brain development.

Mifepristone (RU-486) is a glucocorticoid receptor (GR) antagonist [21]. As an anti-progesterone drug, RU-486 seems to be a very interesting molecule to induce an early vaginal delivery. RU-486 is used in pharmacological abortion, cervical dilatation prior to surgical abortion, and to induce delivery in case of in utero fetal death. Some practicians also use RU-486 to induce labor for late pregnancy with living fetuses; however, this procedure is not commonly used [23]. A pharmacological compound such as RU-486, that is found both in maternal and fetal circulation [24,25], may alter fetal development. In mice, a study described the induction of preterm delivery after RU-486 exposure, and all the pups were described as alive at E12–14 [21]. In another study, Castillo-Ruiz et al. demonstrated that RU-486 exposure decreased physiological cell death occurring after birth in the nucleus accumbens [26]. However, they did not analyze long-term outcomes for pups born following RU-486 injections.

RU-486 is a small molecule (429.6 Da) that passes to the fetal circulatory system. One study demonstrated that RU-486 intraperitoneal (i.p.) administration can activate a viral gene expression reporter within the brain [27], suggesting that it passes through the blood-brain barrier (BBB). Data demonstrate that microglia express some GR [28]; therefore, RU-486 could directly influence microglia reactivity toward a pro- or an anti-inflammatory phenotype. One study demonstrated that RU-486 i.p. administration prior to LPS injection increased neuronal degeneration, and this was worsened in animals with microglial GR deficiency [29]. Another study demonstrated that RU-486 administration at E15 via s.c. injection into the neck region of the dams (to induce preterm birth by cervical ripening [30]), did not modulate pro-inflammatory cytokine levels in fetal brains [31]. However, they did not analyze the long-term consequences of the RU-486 exposition. Finally, regarding the correlation of mouse brain development to human brain development, the induction of PTB by RU-486 at E15 mimics induced PTB occurring during the second trimester of pregnancy. There is no evidence of the long-term impact of RU-486 administration that mimics induced PTB in the third trimester of pregnancy when newborns can be resuscitated; some will present PWMIs.

In our study, in vivo, we evaluated RU-486 subcutaneous (s.c.) administration at E18.5 in pregnant mice to induce PTB during this period of PWMI vulnerability. We also evaluated how labor induction by RU-486 modulates microglial reactivity and PWMIs due to IL-1β injection. Moreover, we evaluated in vitro microglial reactivity after a direct exposition to RU-486.

## 2. Results

### 2.1. Effect of RU-486 In Vivo on Dam Survival, Weight, and Juvenile Behavior

To evaluate the after-birth impact of RU-486 labor induction, we administered RU-486 s.c. at E18.5; this induced a vaginal delivery 24 h after [21,26]. Control animals (referred to as SHAM) were delivered at E20. In pups delivered from RU-486 labor induction, we also wanted to evaluate how perinatal inflammation induced by repeated IL-1β injections [6] could impact development (the in vivo protocol is described in Figure 1A).

We evaluated the survival rate and animals’ weight at postnatal day (P)8. Perinatal exposure to IL-1β induced a 2.5% rate of mortality compared to PBS-exposed animals. Labor induction by RU-486 induced 7.5% mortality both in PBS- and IL-1β-exposed animals (Figure 1B); it was not significant. IL-1β-exposed animals were significantly lighter than PBS-exposed animals following both SHAM and RU-486 labor induction (Figure 1C).

As previously described [16], we evaluated juvenile communication and social behavior. Ultrasonic vocalizations (USVs) are the common rodent communication method (reviewed in [32]). Previously, our team demonstrated that following IL-1β exposure, there is a decrease in USV emissions at P2. Consistent with these data, we demonstrated here that IL-1β-exposed animals emitted significantly fewer calls, regardless of RU-486 labor induction (or in both SHAM and RU-486 labor induction groups, Figure 1D). Mean call duration and frequency were not modulated either by RU-486 or IL-1β. Juvenile social behavior was evaluated by the nest odor preference test. This test evaluates pups’ preference toward a zone enriched with nest sawdust with potential social interaction. We demonstrated that SHAM/IL-1β-exposed pups spent less time in the nest zone in comparison to SHAM/PBS-exposed pups; such differences were not found in animals born from RU-486 labor induction. Moreover, the latency to first enter the nest zone was significantly higher in SHAM/IL-1β-exposed animals, and again this was not found in animals born from RU-486 labor induction (Figure 1E).

Taken together, we demonstrated that, at P8, RU-486 labor induction impacted pups’ mortality at a rate under 10%, without affecting pups’ body weight both in the control and perinatal inflammation conditions. Similarly, RU-486 labor induction did not modulate juvenile behavior deficits associated with IL-1β-exposure.

### 2.2. Effect of RU-486 on Microglial Reactivity In Vivo

Microglial reactivity is a hallmark of PWMIs associated with preterm birth both in humans and in mice [15,16,33]. Therefore, we evaluated the impact of RU-486 labor induction on microglial reactivity both in the control and perinatal inflammation conditions.

Fluorescence-activated cell sorting (FACS) has been described as a relevant method for evaluating microglial reactivity [34]. Microglia are defined by FACS as CD11B^+^CD45^low^ cells, whereas infiltrating macrophages are defined as CD11B^+^CD45^high^ cells [35]. Microglial reactivity can be evaluated by measuring the mean fluorescence intensity (MFI) of CD11B [36] and CD18 [37]. At P3, IL-1β-exposed animals presented a significant increase in the percentage of CD11B^+^CD45^high^ cells both in animals born spontaneously and following RU-486 labor induction (at 0.87 ± 0.1% and 1.22 ± 0.2%, respectively, Figure 2A). CD11B^+^CD45^high^ cells from SHAM/IL-1β-exposed animals showed a significantly increased expression of CD11B (MFI at 63,285 ± 1150) that was significantly downregulated in RU-486/IL-1β-exposed animals (MFI at 55,309 ± 2617, Figure 2B,C). At P3, the percentage of CD11B^+^CD45^low^ cells was not modulated either by RU-486 labor induction or the perinatal administration of IL-1β (Figure 2D). However, CD11B^+^CD45^low^ cells from IL-1β-exposed animals showed a significant increase in CD11B MFI both in spontaneous and RU-486-induced deliveries (Figure 2E,F). Similarly, the percentage of CD11B^+^CD45^low^ cells expressing CD18 was not impacted either by RU-486 or IL-1β (Figure 2G); however, CD18 MFI was significantly increased in SHAM/IL-1β-exposed animals (MFI at 5237 ± 287.3). This induction was also found in RU-486/IL-1β-exposed animals, but to a significantly lower extent (MFI at 4563 ± 133.2, Figure 2H,I). C-X3-C motif chemokine receptor 1 (CX3CR1) is generally used as a marker for mature microglia. During development, CX3CR1 is associated with the microglial synaptic pruning function [38]. Here, we assessed CX3CR1 expression to evaluate whether microglia exposed to RU-486 and/or IL-1β deviated from their normal developmental function. The percentage of CD11B^+^CD45^low^ cells expressing CX3CR1 was not impacted either by RU-486 labor induction or the perinatal administration of IL-1β. However, in both delivery modes, IL-1β-exposed animals presented a significantly lower CX3CR1 MFI (SHAM and RU-486, respectively, at 1668 ± 45.24 and 1507 ± 115.7, Figure 2K,L).

Microglia can adopt a large spectrum of reactivity phenotypes. Our team classifies microglial reactivity into three main transcriptomic profiles: (i) pro-inflammatory, which is neurotoxic and characterized by the expression of *tumor necrosis factor* (*Tnf* mRNA), *prostaglandin-endoperoxide synthase 2* (*Ptgs2* mRNA), and *cluster of differentiation* (*Cd)32* mRNA); (ii) immunoregulatory, characterized by the expression of *interleukin 1 receptor antagonist* (*Il1ra*) mRNA, *suppressor of cytokine signaling 3* (*Socs3*) mRNA, and *interleukin 4 receptor alpha* (*Il4ra*) mRNA; and (iii) anti-inflammatory, that is, neuroprotective and characterized by *insulin-like growth factor 1* (*Igf1*) mRNA and *galectin-3* (*Lgals3*) mRNA [15,16,39]. Another readout that can be used to evaluate microglial reactivity in vitro and in vivo is to quantify these markers’ mRNA expression (Figure 3A–C). As previously described [15,16], perinatal exposure to IL-1β in spontaneously delivered pups induced a significant increase in pro-inflammatory (*Tnf*, *Cd32*, and *Ptgs2* mRNA fold changes, respectively, at 22.89 ± 2.03, 2.06 ± 0.18 and 22.11 ± 2.21, Figure 3A) and immune-regulatory markers (*Il1rn*, *Socs3*, and *Il4ra* mRNA fold changes, respectively, at 31.48 ± 4.63, 21.64 ± 2 and 18.94 ± 1.76, Figure 3B). Pups born from RU-486-induced labor and administered PBS presented a significant microglial upregulation of *Tnf* mRNA (fold change at 14.95 ± 4.14), *Socs3* and *Il4ra* mRNA (fold changes at 15.82 ± 4.31 and 16.33 ± 4.7, respectively), and *Igf1* mRNA (fold change at 2.65 ± 0.66, Figure 3C). RU-486/IL-1β-exposed animals did not up-regulate the expression of these markers in comparison to RU-486/PBS-exposed animals. However, RU-486/IL-1β-exposed animals significantly downregulated *Il1rn* mRNA in comparison to SHAM/IL-1β-exposed animals (fold change at 13.96 ± 1.87).

Reactive oxygen species (ROS) are toxic compounds generated by inflammatory and/or dying cells. Microglia can produce ROS through their membrane NADPH oxidase (reviewed in [40]), and our team previously demonstrated that ROS production can be an easy functional readout for microglial reactivity [16]. SHAM/IL-1β-exposed animals produced significantly more ROS than SHAM/PBS-exposed animals (fold change at 14.69 ± 3.87), as previously described [16]. RU-486/IL-1β-exposed animals also presented a significant increase in ROS production by microglial cells, but to a significantly lower extent than SHAM/IL-1β-exposed animals (fold change at 7.97 ± 0.67, Figure 3D).

Altogether, these data demonstrated that even if microglial reactivity is induced by perinatal inflammation, this reactivity was lower in animals born by RU-486-induced labor in comparison to animals born spontaneously.

### 2.3. RU-486 Impacts on Oligodendrocyte Differentiation and Subsequent Myelination

PWMIs have multiple causes, but an emerging consensus is that PWMIs are due to an arrest of OL differentiation at an OPC stage when they are not able to produce myelin. Previously, our team demonstrated in our perinatal inflammation model that (i) there was an imbalance of platelet-derived growth factor receptor (PDGFR)α (OPC stage) and Oligodendrocyte Marker (O)4 (pre-OL stage that started to myelinate); (ii) this induced a decrease in myelin protein gene expression; leading to (iii) the decreased expression of myelin proteins [6,15,16].

We first evaluated by FACS the percentage of PDGFRα^+^ and O4^+^ cells at P3 (Figure 4A,B). We demonstrated that IL-1β induced a significant decrease in PDGFRα+/O4+ cells and O4^+^ population independent of delivery mode (at 1.11 ± 0.05% and 3.13 ± 0.24%, respectively, Figure 4B). Interestingly, in RU-486/PBS-exposed animals, we saw a significant increase in O4^+^ cells at P3 (5.87 ± 0.32%). This can be correlated to the microglial expression of Igf1 mRNA (Figure 4C, see discussion below).

We then evaluated, at P22, the following myelin protein gene expressions: *myelin basic protein* (*Mbp*), *myelin oligodendrocyte glycoprotein* (*Mog*), *myelin-associated oligodendrocyte basic protein* (*Mobp*), and *proteolipid protein 1*(*Plp*) mRNA. In both delivery mode groups, IL-1β-exposed animals showed a significant decrease in all myelin protein gene expression (Figure 4C). To confirm whether that transcriptomic profile also holds true at the protein level, we performed immunohistochemistry on P22 brain slices against MBP (Figure 4D,F). SHAM/IL-1β-exposed animals expressed significantly less MBP in the motor cortex and in the anterior corpus callosum (fold changes at 0.8 ± 0.06 and 0.72 ± 0.09, respectively). RU-486/IL-1β-exposed animals expressed significantly less MBP only in the motor cortex (fold change at 0.84 ± 0.07, Figure 4E,G).

Altogether, we can conclude that RU-486-induced labor by itself did not induce PWMIs, however, combined with perinatal inflammation, PWMIs occurred due to the diminution of myelin protein expression.

### 2.4. RU-486 Increased Microglia Phagocytosis In Vitro

Microglial cells are the main actors of neuroinflammation in the case of PWMIs [11,15,16]. Since microglial cells express glucocorticoid receptors and given the fact that RU-486 may penetrate the BBB [27], we evaluated the direct effect of RU-486 on a primary microglial culture as previously described [16]. Cells were incubated with RU-486 for 1 h prior to a 3-h stimulation with IL-1β and interferon (IFN)-γ. The lysosomal membrane protein (LAMP)-1 is a glycoprotein highly expressed in lysosomes [41]. An increasing amount of LAMP-1 in microglia (labeled using ionized calcium-binding adapter molecule 1 (IBA-1)) can be associated with increasing microglial reactivity (Figure 5A). IL-1β+IFN-γ-stimulated SHAM IBA-1^+^ cells showed a significant increase in LAMP-1 immunoreactivity (integrated density at 3508 ± 189.6). RU-486 pre-exposition significantly increased this response to IL-1β+IFN-γ (integrated density at 5699 ± 338.7, Figure 5B).

We then evaluated the microglial phagocytic activity by exposing cultures to Cy3 beads. Reactive microglia inclined toward a pro-inflammatory phenotype show an increase in the amount of Cy3 bead intracellularly, which can be interpreted as increased phagocytic activity [42] (Figure 5C). As expected, under IL-1β + IFN-γ stimulation, SHAM IBA-1^+^ cells showed a significant increase in fluorescence (integrated density at 7283 ± 335.1, Figure 5D). Microglial pre-treatment with RU-486 significantly increased fluorescent bead content both in PBS- and IL-1β+IFN-γ-stimulated conditions (integrated density at 9148 ± 415.1 and 9103 ± 273.3, respectively, Figure 5A).

As in the brain extracts, we also evaluated by RT-qPCR the expression of the microglial reactivity marker in microglial cultures. IL-1β+IFN-γ-stimulated cells increased significantly in *Tnf*, *Cd32*, *Ptgs2*, *Il1rn*, *Socs3*, and *Il4ra* mRNA expression. These expression profiles were similar in RU-486 pre-treated cells (Figure 6A,B). IL-1β+IFN-γ stimulation did not modulate anti-inflammatory marker expression in either SHAM or RU-486 conditions (Figure 6C).

In summary, direct microglial exposition to RU-486 in vitro systematically potentiated microglial reactivity after exposition to a pro-inflammatory stimulus; this was considerably different from what we observed in vivo.

## 3. Discussion

Preterm birth represents 1 out of 10 births worldwide and is the leading cause of death before the age of five [1]. Some survivors present white matter injuries due to an arrest of oligodendrocyte maturation following microglial reactivity associated with perinatal inflammation. NDDs are linked to these white and/or grey matter injuries associated with preterm birth. Recently, Canini et al. described a cognitive development blueprint by MRI as a consistent functional maturation of subcortical networks in the fetal brain that could be altered in the case of NDDs [43]. One hypothesis regarding the correlation between white matter and gray matter injury is that WMI may activate astrogliosis and axonal myelination defects in preterm brains associated with neuroinflammation, in part due to the blockade of pre-OL maturation. These axonal defects lead to grey matter injuries due to abnormal axonal physical tension [4].

There is currently no recommendation concerning the mode of delivery in case of preterm birth. This is mainly because there is a lack of data in the literature regarding the impact of the delivery mode on brain development. Mifepristone (RU-486) is one way to pharmacologically induce delivery both in humans [23] and rodents [21]. Our present studies aimed to evaluate, in mice, (i) whether RU-486-induced labor impacted microglial reactivity and oligodendrocyte maturation, and (ii) whether RU-486-induced labor modulated the same factors in a context of perinatal inflammation by IL-1β. In this study, we demonstrated that RU-486 leads to labor induction in 100% of cases. The communication and social behavior of pups born after RU-486-induced labor were not disturbed. Communication deficits due to IL-1β administration were equal in both delivery modes. Microglial reactivity was not modulated by RU-486-induced labor; however, when combined with perinatal inflammation, RU-486-induced labor minimized microglial reactivity. Similarly, RU-486-induced labor alone did not impact oligodendrocyte maturation or the subsequent myelination process. The significant microglial reactivity decrease observed following RU-486-induced labor did not prevent PWMIs due to perinatal inflammation by IL-1β. In vitro, stimulated microglia were subject to an RU-486 exposition: RU-486 significantly increased microglial reactivity, suggesting that other cellular actors might also participate in vivo.

Other risk factors of preterm birth independent of premature delivery itself may act on neurodevelopmental outcomes. For example, preterm birth can frequently be associated with intrauterine infection leading to microglial activation and hypomyelination. Other causes of preterm birth, such as placental impairments, can provoke cerebral lesions due to hypoxic-ischemic encephalopathy. A recent article proposed a model of assessing preterm birth risk factors to be used in clinical practice using relevant intrauterine and extrauterine factors [44]. In the future, the development of such a model for NDDs according to the preterm birth etiology may be relevant.

In our study, we observed a mortality rate increase in the case of RU-486-induced labor. RU-486 labor induction induced a modification of maternal behavior. Indeed, some of the RU-486-injected mothers had difficulties taking care of the newborns immediately after birth, although systematic adoption was performed as soon as possible. Roe et al. demonstrated recently that RU-486 increases maternal abdominal pain even though it decreases the total duration of miscarriage management [45]. We hypothesized that the maternal impairment observed in mice right after delivery was due to abdominal pain linked to RU-486 administration. This maternal behavioral impairment was not described in any previous article concerning animal models with RU-486.

Previously, in a human randomized trial, RU-486 administrated to the mother was found in the fetal circulatory system within a few hours, indicating that RU-486 rapidly crosses the placenta [24]. RU-486 is also known to cross the blood-brain barrier because of its amphiphilic steroid properties [46] and to be a progesterone receptor (PR) and glucocorticoid receptor (GR) antagonist. Microglia express the GR but do not express the PR [28], suggesting that the potential effects of RU-486 on microglia are mediated through the GR. The Hypothalamic-pituitary-adrenal (HPA) axis is involved in the physiological response to environmental or endogenous stress, responsible for producing the glucocorticoid (GC) hormone in the adrenal gland [47]. This system is one of the main actors of homeostasis regulation. Microglial activation can stimulate the HPA axis, and high levels of GC can activate microglia [48]. GC seem to have both pro-inflammatory and anti-inflammatory functions; by extension, its GR antagonist RU-486 does too [48]. On the one hand, Carrillo-de Sauvage et al. showed that, following LPS administration, RU-486 increased neuronal degeneration by microglial GR inhibition [29]. On the other hand, RU-486 has been used in many studies as a “non-infectious model” of preterm birth in contrast to LPS-induced labor. Burd et al. compared LPS intrauterine injection with RU-486 i.p. injection at E15 to induce labor. Only the LPS labor induction resulted in an increase in brain cytokine mRNA and disrupted fetal neuronal morphology [31]; RU-486 labor induction did not impact neuro-inflammation or its secondary brain lesions at E15. Frank et al. showed that RU-486 suppressed stress-induced glucocorticoid signaling and blocked the stress-induced sensitization of the pro-inflammatory response of microglia [49]. Another study showed that the ablation of GC signaling with RU-486 was found to hinder the LPS-mediated expression of various immune genes such as IL-1β [50]. This is consistent with our results, in which we found a lower level of microglial reactivity induced by inflammation in animals born by RU-486-induced labor.

RU-486 seemed to have a direct link to the mobilization and differentiation of oligodendrocyte precursor cells and to the production of myelin by influencing some glucocorticoid-dependent neuroinflammatory factors such as oncostatin M [50]. It is now well accepted that the use of GC could provoke neurodevelopmental delays in preterm children [51]. A study demonstrated that high doses of GC treatment for prematurely delivered rabbit pups arrested the maturation of OL and MBP expression. In that study, associated RU-486 treatment reversed this effect by blocking GR [52]. Indeed, GR are expressed in the cytoplasm of OL [53]. The last days of pregnancy and maternal stress at delivery expose the fetus to increased levels of glucocorticoids [54]. The RU-486 block of OL GR just before delivery could participate in the significant increase in mature OL precursors (O4^+^ cells) at P3 observed in our study in the case of RU-486-induced labor without any context of inflammation. The RU-486 increase in mature OL precursors could also be linked to the observed significant increase of IGF1 in the case of exposition to RU-486 without inflammation. We know that IGF1 is necessary for normal oligodendrocyte development and myelination [55]. However, this small positive effect of RU-486 seems not to be sufficient to significantly increase MPB expression or reverse the arrest of oligodendrocyte precursor maturation in the case of exposition to IL-1β.

One of the main well-known functions of microglia during development is the modulation and elimination of neuronal synapses and myelin by phagocytosis [56,57]. Horchar et al. showed that microglia phagocytosis properties were exacerbated in the case of chronic stress-induced glucocorticoid signaling and that RU-486 administration reduced the markers of microglia phagocytosis [58]. In another study in mice, cold stress reduced the phagocytosis of microglia, which was not reversed after RU-486 administration [59]. These results are heterogeneous and not consistent with our results ex vivo that observe an increase in microglial phagocytosis properties in PBS and IL-1β conditions. The observed potentiated microglial reactivity ex vivo in opposition to the one observed in vivo may be explained by the toxicity of an RU-486 dose used directly on microglia in our experiment. In the case of maternal injection of RU-486 and transplacental passage, the circulating fetal dose of RU-486 should be lower.

In summary, RU-486 seems to be a clinically relevant model for inducing PTB during the period of EoP vulnerability, with an acceptable mortality rate. RU-486-induced birth animals seem to have a comparable outcome in terms of weight and juvenile behavior. RU-486-induced birth did not induce PWMIs by itself and did not restore inflammation-induced PWMIs despite a decrease in inflammation-induced microglia reactivity and an increase in mature OL precursor cells.

## 4. Materials and Methods

### 4.1. Animal Handling, Labor Induction by RU-486, and EoP Model

Experimental protocols were approved by the Ethics Committee and the services of the French Ministry in Charge of Higher Education and Research (#10469-2017070312302631). Experiments were performed on OF1 strain mice (Charles River, France). Naïve pregnant mice were injected subcutaneously with 10^−2^ M of RU-486 (Sigma-Aldrich, St. Louis, MI, USA) dissolved in 1 mL of phosphate-buffered saline (PBS) 1× or with 1 mL of PBS 1× alone (referred to as Sham animals) at E18.5. Delivery occurred at E19.5 for RU-486-administered animals and at E20 for Sham animals. Male pups delivered by these mothers were injected intra-peritoneally (i.p.) twice a day from P1 to P5 with 10 μg/kg of IL-1 β (Miltenyi Biotec, Bergisch Gladbach, Germany) diluted in 5 μL of PBS 1× or with 5 μL of PBS 1× alone (referred to as PBS animals). Pups were monitored twice a day for any sign of distress according to a clinical score, including feeding, respiratory rates, and weight gain.

### 4.2. Brain Collection, Dissociation, and Microglia Magnetic Cell Sorting

For the ex vivo microglia culture, P8 naïve animals were decapitated. For cell sorting (P3) and flow cytometry (FACS, P5) analysis, pups were injected with an overdose of Euthazol and perfused intracardially with 0.9% NaCl. Brains without the cerebellum and olfactory bulbs were collected and dissociated using the Neural Tissue Dissociation Kit containing papain and the gentleMACS Octo Dissociator with Heaters. Cells for FACS analysis were collected after rinsing (see below). Magnetic beads coupled with mouse anti-CD11B antibodies (microglia) were used for cell isolation according to the manufacturer’s protocol (Miltenyi Biotec, Germany) as previously described [15,16]. Brains for myelin protein gene expression were collected after animals’ dislocation and stored at −80 °C prior to mRNA extraction.

### 4.3. Ex Vivo Microglia Cell Culture

Following cell sorting, CD11b+ cells were suspended in Macrophage-Serum free Media (SFM Media, Gibco, Waltham, MA, USA) with 1% penicillin/streptomycin (Gibco, Waltham, MA, USA) at 6.10^5^ cells/mL concentration (0 days in vitro (DIV)). We used 12-well culture plates and µ-Slide 8 Well glass Bottom (Ibidi, Gräfelfing, Germany) containing 1 mL/well and 250 µL/well of cell suspension, respectively. The media were changed at 1 DIV. At 2 DIV, microglia cells were stimulated with RU-486 (Sigma-Aldrich, MI, USA) at 10^−3^ M concentration diluted in absolute ethanol. Group control was stimulated only by absolute ethanol at the same concentration. After 1 h of RU-486 incubation, IL-1β (50 ng/mL) and interferon (IFN)-γ (20 ng/mL) were added to the culture media (PBS for the control group). After 3 h of stimulation, the media were aspirated, and the 12-well plates were kept at −80 °C prior to mRNA extraction. The µ-slides were fixed at room temperature with 4% formaldehyde for 20 min.

### 4.4. RNA Extraction and Real-Time qPCR

mRNA from both CD11B+ cells at P3 and ex vivo microglial cultures at P8 were extracted by the RNA XS Plus (Macherey-Nagel, Düren, Germany) according to the manufacturer’s protocol. Total RNA from P22 brains was extracted using a NucleoSpin RNA extraction kit (Macherey-Nagel, Germany) according to the manufacturer’s protocol. Extracted mRNA were analyzed for quality and concentration by spectrophotometry (Nanodrop^®^2000 (ThermoFisher Scientific, Waltham, MA, USA). Reverse transcription was performed at P3/8 and P22 using 350 ng and 1000 ng of mRNA, respectively, using an iScript cDNA synthesis kit (BioRad, Hercules, CA, USA). Real-time quantitative PCRs (RT-qPCR) were performed by triplicate sample using SYBR Green Supermix (BioRad, CA, USA) for 44 cycles (denaturation 5 s at 96 °C, hybridization 10 s at 60 °C). Melting curve analyses allowed for specific amplification evaluation. The design of the primers was performed using Primer3 plus or PrimerBLAST software (Table 1). mRNA levels were calculated using the 2 delta Ct method after normalization with *Rpl13a* mRNA as the reference mRNA; relative expression is expressed in comparison with SHAM/PBS-exposed samples as previously described [15,16,33].

### 4.5. Measurements of Reactive Oxygen Species Production by Luminometry

After CD11B+ cell magnetic sorting, 20,000 cells of each sample were suspended in Hank Balanced Salt Solution with Ca^2+^ and Mg^2+^ (HBSS^+/+^) and incubated with luminol (50 µM; Sigma) at 37 °C in the dark for 10 min. CD11+ cells were stimulated with phorbol 12-myristate 13-acetate (PMA, Sigma-Aldrich, MI, USA) to enhance basal ROS production [40,60]. Reactive oxygen species (ROS) productions were assayed in duplicate. Analysis was done using a luminometer (CentroLB 960; Berthold Technologies, Bad Wildbad, Germany) in a 96-well plate. The signal was recorded for 1 s every 2 min for 20 min. Results were analyzed from the area under the curve (AUC) of luminescence production. Relative expression is expressed in comparison with SHAM/PBS-exposed samples.

### 4.6. Flow Cytometry Analyses

After brain dissociation, as described previously, the cellular count was performed for each sample (Nucleocounter, Chemometec, Denmark) and suspended in Dulbeccos’s Phosphate Buffer Saline (Gibco, MA, USA), 2 mM EDTA (Sigma-Aldrich, MI, USA), 0.5% Bovine Serum Albumin (Miltenyi Biotec, Germany, referred to as FACS buffer) at 10^7^ cells/mL. Immunostaining was performed: anti-CD11b BV421, anti-CD45 BV510, anti-CX3CR1 PE Cy7 (Sony Biotechnology, San Jose, CA, USA), anti-CD18-APC (BD Biosciences, East Rutherford, NJ, USA) for microglia phenotyping, and anti-O4 Vio Bright B515 and anti-PDGFRα PE (Miltenyi Biotec, Germany) for oligodendrocyte phenotyping. Samples were kept at 4 °C in the dark overnight, and flow cytometry analysis was performed the following day (LSR FortessaTM X-20, BD Biosciences, NJ, USA). Microglia cells were defined as CD11b^+^/CD45^low^ live cells. CX3CR1 and CD18 results were expressed as a percentage of positive cells and mean fluorescence intensity (MFI). Oligodendrocyte cells were defined as O4^+^ PDGFRα^−^, O4^−^ PDGFRα^+^, and O4^+^PDGFRα^+^. O4 and PDGRα MFI were analyzed in the three populations.

### 4.7. Immunofluorescence

Immunofluorescence staining was performed on Ibidi chambers using goat anti-IBA1 (Abcam, Ab507, 1/500), rabbit anti-LAMP1 (Sigma L1418-200UL, 1/150) antibodies, and fluorescent conjugated goat IgG secondary antibodies. For phagocytic assay latex beads, amine-modified polystyrene, fluorescent red (Sigma-Aldrich, MI, USA) was incubated for 1 h prior to fixing Ibidi chambers. Slide reading was performed on a Leica TCS SP8 confocal microscope. Immunofluorescence was calculated in each cell individually using the integrated density value from NIH Fiji software [61].

### 4.8. Immunohistochemistry

At P22, brains were processed into paraffin sections by immediate immersion in 4% formaldehyde for 4 days at room temperature prior to dehydration and paraffin embedding. Sections were realized using a microtome. Immunostaining was performed using a Leica Bond max robot using a BOND Polymer Refine Detection kit and mouse antibody to anti-MBP (Millipore, MAB382, 1/500). Images were acquired using a Nanozoomer slide scanner (Hamamatsu). Images were extracted for each region of interest using NDP View software (Hamamatsu). The intensity of the MBP immunostaining in the anterior corpus callosum and in the motor cortex was evaluated by a densitometry analysis through NIH ImageJ Software. Optical density was deduced from grey scale standardized to the photomicrograph background. One measurement was taken per section (on 40,000 mm^2^ area), and a total of four sections were analyzed in each brain. Relative expression is expressed in comparison with SHAM/PBS-exposed samples.

### 4.9. Behavioral Tests

Ultrasonic vocalizations were recorded, for 3 min at P2, with an ultrasound microphone (Noldus), sensitive to frequencies of 30–90 kHz. A pup, isolated from the litter, was placed into a container (H 4.5 cm × L 10 cm × l 10 cm) inside of a soundproof polystyrene box (H 23 cm × L 37 cm × l 24 cm) to avoid interference from external noises. The box temperature was 23.2 degrees. The microphone was placed 10 cm above the pup’s head. Ultrasonic vocalizations were recorded with the Ultravox XT3.1 software and analyzed using [62].

Social behavior was evaluated by the nest odor preference test at P8 as previously described [16]. The temperature of the room was 24 °C. The measuring platform (L 20 cm × l 13 cm) was composed of three zones: the nest zone, the neutral zone without sawdust, and the clean zone. The sawdust zones and neutral zone measured, respectively, L 7 cm × l 13 cm and L 6 cm × l 13 cm. The pup, isolated from its littermates, was placed in the “neutral” zone of the measuring platform. The time spent in each area was measured. Between the testing of each littermate, the measuring platform and the sawdust zones were cleaned with ethanol 70%.

### 4.10. Statistical Analysis

Data were expressed as mean values with standard error of the mean (SEM). Using GraphPad Prism Software, multiple comparisons in the same data set were analyzed by 1-way analysis of variance (ANOVA) or 2-way ANOVA with the Sidak multiple comparison test or multiple *t*-test.

## Figures and Tables

**Figure 1 ijms-23-04867-f001:**
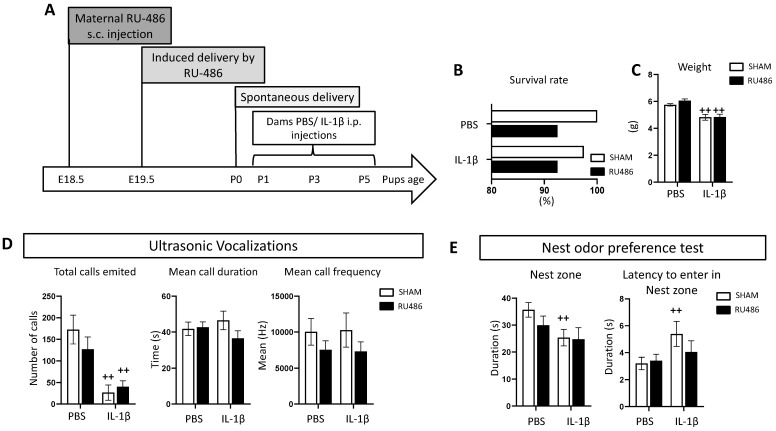
Impact of RU-486 on mortality, weight, and early behavior. (**A**) Schematic timing representation of the double-hit model of RU-486-induced delivery ± perinatal inflammation (IL-1β injections). (**B**) Representation of survival rate (%) at P8 according to the mode of delivery and perinatal inflammation (*n* = 50/group). (**C**) Mean weight (g) at P8 according to the mode of delivery and perinatal inflammation. (**D**) Ultrasonic vocalization test evaluating total calls emitted, mean call duration (s), and frequency (Hz) in SHAM vs. RU-486 animals and PBS vs. IL-1β animals (*n* = 15/group). (**E**) Nest odor preference test at P8 evaluating maternal recognition. Results are presented as mean time spent in nest zone and mean latency time to enter the nest zone (*n* = 15–20/group). Two-way analysis of variance (ANOVA) corrected by the Sidak test (++ *p* ≤ 0.01 comparing IL-1β vs. PBS).

**Figure 2 ijms-23-04867-f002:**
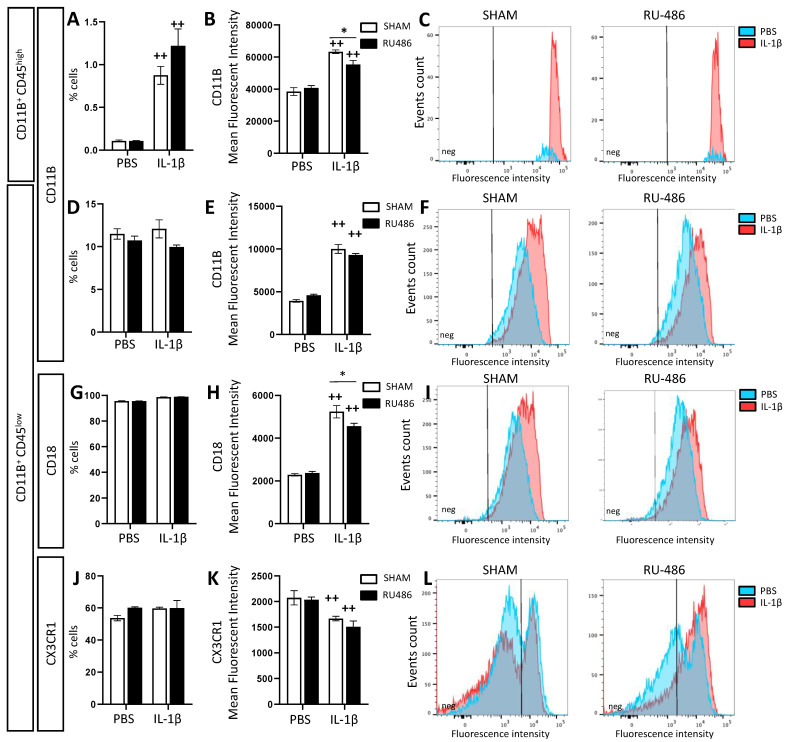
Flow cytometry analysis of the impact of RU-486 on microglial reactivity at P3. Flow cytometry analysis of microglial reactivity at P3 in mice with spontaneous birth or RU-486-induced labor followed by the perinatal administration of PBS or IL-1β. Results are presented in two populations: CD11b^+^/CD45^high^ (**A**–**C**) and CD11b^+^/CD45^low^ (**D**–**L**). The percentages of each population are presented in (**A**,**D**), respectively. In each population, mean fluorescent intensities (MFI) were evaluated for CD11B (**B**,**E**), CD18 (**H**), and CX3CR1 (**K**). For each marker, a representative image of MFI was selected (**C**,**F**,**I**,**L**). Two-way analysis of variance (ANOVA) corrected by the Sidak test (*n* = 6/group, mean SEM), (* *p* ≤ 0.05 comparing RU-486-induced delivery vs. spontaneous delivery, ++ *p* ≤ 0.01 comparing IL-1β vs. PBS).

**Figure 3 ijms-23-04867-f003:**
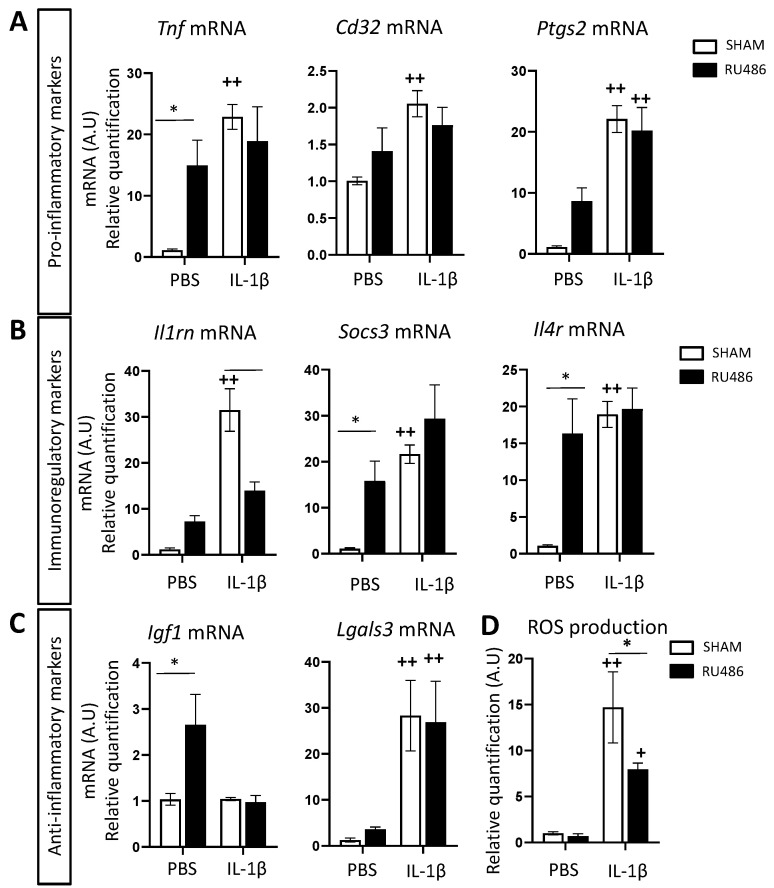
RU-486 impact on ex vivo microglial reactivity marker expression at P3. The relative expression of mRNA encoding pro-inflammatory (**A**), immuno-regulatory (**B**), and anti-inflammatory markers (**C**) of microglia from P3 in mice with spontaneous birth or RU-486-induced labor followed by perinatal administration of PBS or IL-1β. Cells were obtained after CD11B^+^ cell magnetic sorting. Results of RT-qPCR are presented as fold change adjusted to SHAM-PBS-exposed animals. (**D**) Reactive oxygen species (ROS) production (*n* = 6/group, mean SEM) after PMA stimulation, by isolated CD11B^+^ cells from P3 in mice with spontaneous birth or RU-486-induced labor followed by perinatal administration of PBS or IL-1β (*n* = 6/group, mean SEM). Two-way analysis of variance (ANOVA) corrected by the Sidak test (* *p* ≤ 0.05 comparing RU-486-induced delivery vs. spontaneous delivery, ++ *p* ≤ 0.01 comparing IL-1β vs. PBS).

**Figure 4 ijms-23-04867-f004:**
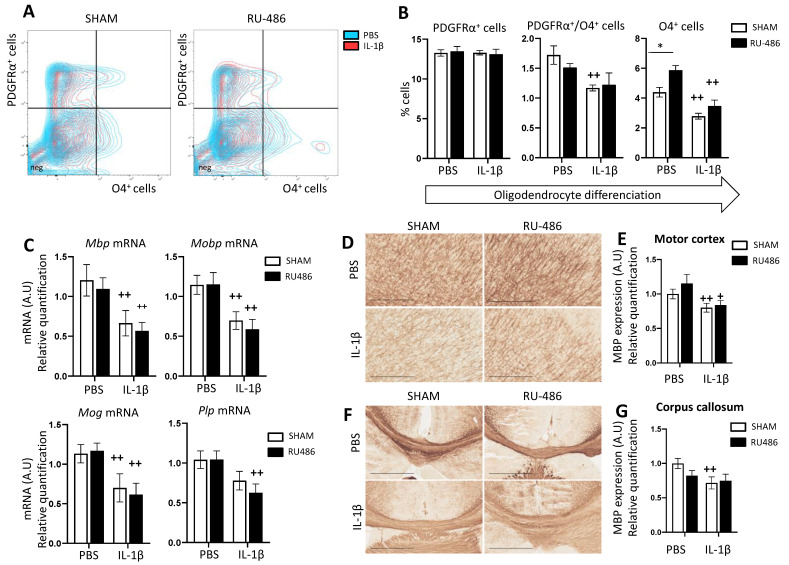
RU-486 impacts on oligodendrocyte differentiation and subsequent myelination. (**A**) Representative contour plot of PDGRα^+^ cells and O4^+^ cells and (**B**) percentage of PDGRα^+^ cells, PDGRα^+^/ O4^+^ cells, and O4^+^ cells from P3 in mice with spontaneous birth or RU-486-induced labor followed by perinatal administration of PBS or IL-1β (*n* = 6/group, mean SEM). Two-way analysis of variance (ANOVA) corrected by the Sidak test (* *p* ≤ 0.05 comparing RU-486-induced delivery vs. spontaneous delivery, ++ *p* ≤ 0.01 comparing IL-1β vs. PBS). (**C**) Relative expression of mRNA-encoding myelin genes in the forebrain at P22 in mice with spontaneous birth or RU-486-induced labor followed by perinatal administration of PBS or IL-1β (*n* = 10/group, mean SEM). Representative image of P22 MBP immunostaining in the motor cortex (**D**) and corpus callosum (**F**) (scale bar 500 µm). (**E**,**G**) Relative quantification of MPB expression at P22 in mice with spontaneous birth or RU-486-induced labor followed by perinatal administration of PBS or IL-1β (*n* = 8–11/group, mean SEM). Multiple t-test (+ *p* ≤ 0.05, ++ *p* ≤ 0.01 comparing IL-1β vs. PBS).

**Figure 5 ijms-23-04867-f005:**
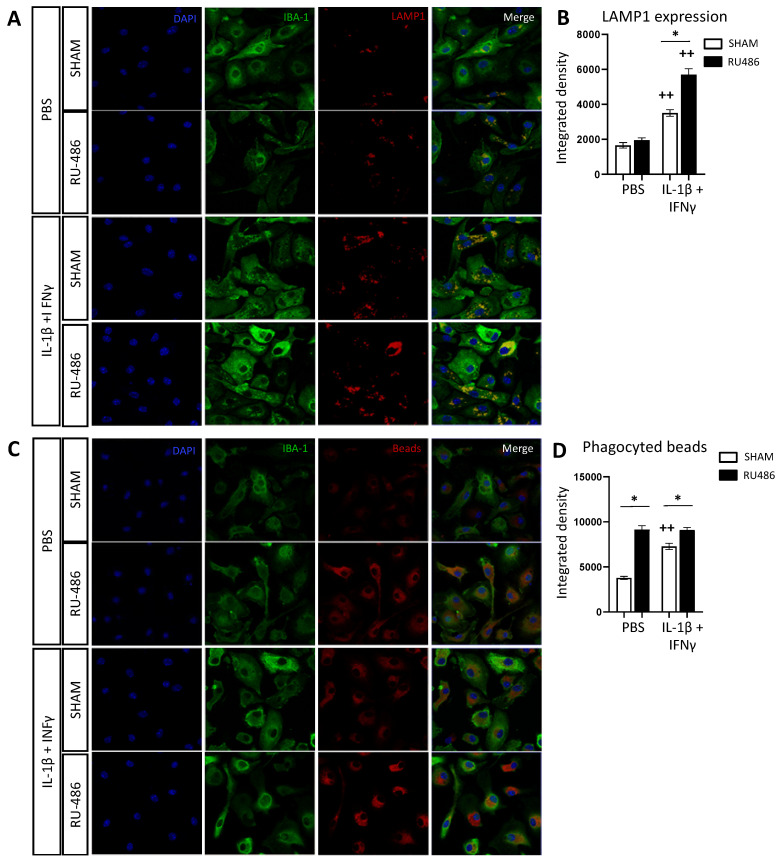
RU-486 increases microglial lysosome LAMP-1 expression and phagocytosis in vitro. CD11B+-cell-sorted microglia were treated with RU-486 at 10^−3^ M diluted in absolute ethanol (or with absolute ethanol only for SHAM group) for 1 h prior to stimulation with IL-1β + IFN-γ or with PBS as the control for 3 h. Immunofluorescence stainings were performed on IBIDI chambers. (**A**) Confocal images of LAMP1 immunofluorescence of in vitro microglial cells (stained with IBA1 antibody) after RU-486 and IL-1β+ IFN-γ exposition. (**B**) Integrated density of LAMP1 immunoreactivity in IBA1+ cells (*n* = 294–582 cells/group). One-way analysis of variance (ANOVA) corrected by the Sidak test (* *p* ≤ 0.05 comparing RU-486 vs. SHAM, ++ *p* ≤ 0.01 comparing IL-1β+ IFN-γ vs. PBS). (**C**) Confocal images of phagocytic assay latex beads immunofluorescence of in vitro microglial cells (stained with IBA1 antibody) after RU-486 and IL-1β+ IFN-γ exposition. (**D**) Integrated density of fluorescence emitted from Cy3 beads in IBA1+ cells exposed to RU486 ± (IL-1β+ IFN-γ) (*n* = 288–626 cells/group). One-way analysis of variance (ANOVA) corrected by the Sidak test (* *p* ≤ 0.05 comparing RU-486 vs. SHAM, ++ *p* ≤ 0.01 comparing IL-1β + IFN-γ vs. PBS).

**Figure 6 ijms-23-04867-f006:**
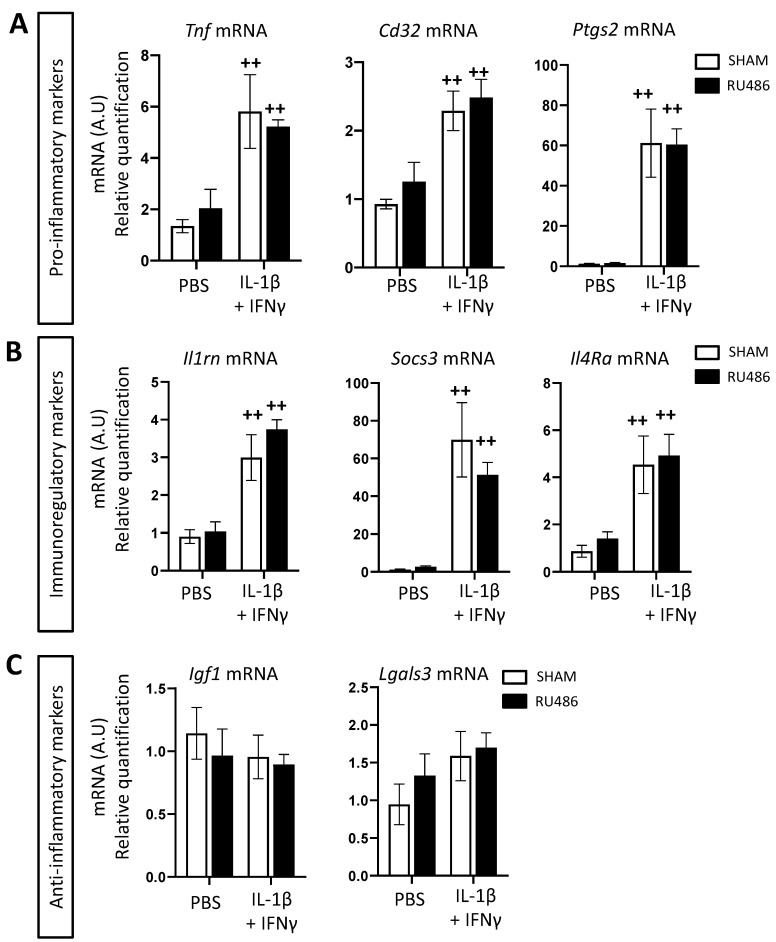
RU-486 exposition did not modulate microglial reactivity in vitro. CD11B+-cell-sorted microglia were treated with RU-486 at 10^−3^ M diluted in absolute ethanol (or with ethanol absolute only for SHAM group) for 1 h prior to stimulation with IL-1β + IFN-γ or with PBS as the control for 3 h. Relative expression of mRNA encoding pro-inflammatory (**A**), immuno-regulatory (**B**), and anti-inflammatory markers (**C**) of microglial reactivity ex vivo. Results of RT-qPCR are presented as fold change relative to the control group SHAM+PBS (*n* = 6 wells/group). Two-way analysis of variance (ANOVA) corrected by the Sidak test (++ *p* ≤ 0.01 comparing IL-1β + IFN-γ vs. PBS).

**Table 1 ijms-23-04867-t001:** Primer sequences for RT-qPCR analysis.

Gene Name	Description	Forward	Reverse
*Rpl13a*	Ribosomal protein L13 a	ACA GCC ACT CTG GAG GAG AA	GAG TCC GTT GGT CTT GAG GA
*Ptgs2*	Prostaglandin-endoperoxide synthase 2 (=Cox2)	TCA TTC ACC AGA CAG ATT GCT	AAG CGT TTG CGG TAC TCA TT
*Tnf*	Tumor necrosis factor	GCC TCT TCT CAT TCC TGC TT	AGG GTC TGG GCC ATA GAA CT
*Il1rn*	Interleukin 1 receptor antagonist	TTG TGC CAA GTC TGG AGA TG	TTC TCA GAG CGG ATG AAG GT
*Il4ra*	Interleukin 4 receptor alpha	GGA TAA GCA GAC CCG AAG C	ACT CTG GAG AGA CTT GGT TGG
*Socs3*	Suppressor of cytokines 3	CGT TGA CAG TCT TCC GAC AA	TAT TCT GGG GGC GAG AAG AT
*Lgals3*	Lectin galactoside-binding soluble 3	GAT CAC AAT CAT GGG CAC AG	ATT GAA GCG GGG GTT AAA GT
*Igf1*	Insulin-like growth factor 1	TGG ATG CTC TTC AGT TCG TG	GCA ACA CTC ATC CAC AAT GC
*Mbp*	Myelin basic protein	CCG GAC CCA AGA TGA AAA C	CTT GGG ATG GAG GTG GTG T
*Mog*	Myelin-oligodendrocyte glycoprotein	AAG AGG CAG CAA TGG AGT TG	GAC CTG CAG GAG GAT
*Plp*	Proteolipid protein	CCA AAT GAC CTT CCA CCT GT	CGA AGT TGT AAG TGG CAG CA
*Mobp*	Myelin-associated oligodendrocyte basic protein	TCCACAGGAAC CTTTCACAA	TCCT GGCCATTTTCTGACT
*Cd32*	Cluster of differentiation 32	CTG GAA GAA GCT GCC AAA AC	CCA ATG CCA AGG GAG ACT AA

## Data Availability

Not applicable.

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
