# Peer review of "The Impact of Mouse Preterm Birth Induction by RU-486 on Microglial Activation and Subsequent Hypomyelination"

_ijms, 2022, doi:10.3390/ijms23094867_

Round 1
Reviewer 1 Report
I find the study interesting and sound. The research question is relevant from a clinical perspective and the are major potentials for clinical translation, as well. However I find that the discussion is somehow weak and I would reccomend some constructive criticisms:
1.The authors state correctly that in the case of preterm birth, an arrest of their differentiation at OL precursors occurs leading to a delay in brain myelinization and periventricular white matter injury.
Recent work described prenatal fingerprints of neurodevelopment in relation to preterm birth (ref 1) Please discuss the issue of specific blueprint of connectivity (ref 1) in relation to their findings of hypomyelination: I believe this is highly relevant and these data must be discussed together. Is altered myelination vs inflammation in the authors opinion, a possible esplanation for specific patterns of connectivity found at resting state fetal MRI shown by reference 1?
Risk factors of preterm birth must be discussed within this model. Is is possible that risk factors themselves may act on the outcome of their study rather that premature delivery itself? In other words is it possible that the outcome preterm birth is indeed as urrogate outcome and the real outcomes are instead all those insults related to etiology of PTB (eg intra amniotic inflammation, placental dysfunction and altyered fetal oxygenation, ets). Please discuss biefly this concept with a sentence including the reference suggested. The reference suggested is proposing a model assessing preterm birth risk factors with machine learning techniques creating novel and echonomical prediction models whch may be well describing phenomena related the the anbornmal neurodevelopment emerging from their study.
I reccomend additional review for lab methodology.
References
1. Matteo Canini, et al. Subcortico-Cortical Functional Connecivity in the Fetal Brain: A Cognitive Development Blueprint, Cerebral Cortex Communications, Volume 1, Issue 1, 2020, tgaa008, https://doi.org/10.1093/texcom/tgaa008
2. Della Rosa PA, et al. A hierarchical procedure to select intrauterine and extrauterine factors for methodological validation of preterm birth risk estimation. BMC Pregnancy Childbirth. 2021 Apr 16;21(1):306. doi: 10.1186/s12884-021-03654-3. PMID: 33863296; PMCID: PMC8052693.
Reviewer 2 Report
I think this is a very good paper and quite interesting . The main question is, can the authors through their work and by presenting more papers recommend a safe way of delivery in PPROM? Moreover, would that differ in cases if inflammation? (that is suggested in the use of mifepristone and IL6.
One question I have is how many mice did the authors use for the study group and how many for the control group?
Minor English mistakes, some of which I point out because they are quite noticeable:
line 150 - to evaluate
line 227 - to many that’s line 229 - you should use ; instead of , 388 - in opposition to (at the opposite?) 412 - preferably the English version - naiveAuthor Response
Please see the attachment.
